# Self-Supervised Iterative Contextual Smoothing
# for Efficient Adversarial Defense against Gray- and Black-Box Attack

**Sungmin Cha** [1 2]   **Naeun Ko** [3]   **Youngjoon Yoo** [2 3]   **Taesup Moon** [1]

## Abstract

We propose a novel and effective input transformation based adversarial defense method against gray- and black-box attack, which is computationally efficient and does *not* require any adversarial training or retraining of a classification model. We first show that a very simple iterative Gaussian smoothing can effectively wash out adversarial noise and achieve substantially high robust accuracy. Based on the observation, we propose Self-Supervised Iterative Contextual Smoothing (SSICS), which aims to reconstruct the original discriminative features from the Gaussian-smoothed image in context-adaptive manner, while still smoothing out the adversarial noise. From the experiments on ImageNet, we show that our SSICS achieves both high standard accuracy and very competitive robust accuracy for the gray- and black-box attacks; *e.g.*, transfer-based PGD-attack and score-based attack. A noteworthy point to stress is that our defense is free of computationally expensive adversarial training, yet, can approach its robust accuracy via input transformation.

## 1. Introduction

Due to the vulnerability of deep neural networks to the adversarial attacks, an "arms race" research between attack and defense methods has been recently pursued very actively. Namely, when a method was proposed to defend against existing attack methods, it soon got broken by stronger new attacks, and vice versa. Among the proposed defense methods, the so-called *adversarial training*, which requires re-training of the networks from scratch to meet the robust performance

criterion, was shown to be the most robust against strong white-, gray- and black-box attacks (Dong et al., 2020). However, the obvious drawback of such defense is that the re-training becomes computationally expensive for large training datasets or models.

Input transformation based *empirical* defense methods (Dziugaite et al., 2016), on the other hand, simply transform the given input data in a way that the adversarial noise can be removed before it gets fed into the network; thus, re-training is not necessary. However, early examples of those defense methods later on were shown to be largely broken by not only adaptive attacks, *e.g.*, (Carlini & Wagner, 2017) and (Madry et al., 2017), but also various kinds of strong gray- and black-box attacks (Guo et al., 2017a; Dong et al., 2020). Another line of work is the *certified* defense (Cohen et al., 2019), which also provides with robustness guarantees, and recently, (Salman et al., 2020) proposed a certified defense method that first denoises the Gaussian noise-corrupted input data before passing it to the classification network. The method can be interpreted as performing a special form of input transformation, hence, did not require a re-training of the network as well. However, while their method can give a certificate on the robust accuracy, its standard accuracy tends to deteriorate significantly (as we show later).

Inspired by (Salman et al., 2020), we propose a new input transformation based empirical defense method that can achieve both strong standard and robust accuracy against gray- and black-box attacks without *any* adversarial training or retraining of a classification model. To that end, we first demonstrate that a very simple *iterative* Gaussian smoothing, which effectively washes out the adversarial noise, can achieve surprisingly high robust accuracy. We then devise Self-Supervised Iterative Contextual Smoothing (SSICS), which aims to iteratively reconstruct the original discriminative features from the Gaussian-smoothed image in a context-adaptive way, while still smoothing out the adversarial noise. To realize such smoothing, we devise a new, efficient version of blind-spot network (BSN) based on a recent work (Byun et al., 2021). As a result, we show that our SSICS, an input transformation-based defense, can almost approach the standard and robust accuracy of the adversarial training-based state-of-the-art defense method on the gray-

[1]Department of Electrical and Computer Engineering, Seoul National University, South Korea [2]NAVER AI Lab, South Korea [3]Face, NAVER Clova, South Korea. Correspondence to: Taesup Moon <tsmoon@snu.ac.kr>.

*Accepted by the ICML 2021 workshop on A Blessing in Disguise: The Prospects and Perils of Adversarial Machine Learning.* Copyright 2021 by the author(s).

and black-box attacked ImageNet validation set.

## 2. Related Work

**Adversarial attack and defense**    After (Szegedy et al., 2013) revealed the vulnerability of neural networks, various papers have proposed different methods to attack the neural network (Madry et al., 2017; Goodfellow et al., 2014; Kurakin et al., 2016; Andriushchenko et al., 2020). Against these attacks, several papers proposed adversarial defense methods categorized by adversarial training (Shafahi et al., 2019; Xie et al., 2019) and input transformation (Guo et al., 2017b; Xu et al., 2017; Dziugaite et al., 2016; Xie et al., 2017). However, input transformation methods are reported to easily be incapacitated by white-box attack (Dong et al., 2020). In more weak attack settings, such as gray- and black-box attacks, (Guo et al., 2017b) suggested that input transformation with additional training achieves a comparative performance against adversarial examples from the attack scenarios. However, these still show weakness in strong gray- and black-box attacks (Dong et al., 2020). Related studies on blind-spot network are listed in Supplementary Materials (SM).

## 3. Method

### 3.1. Preliminary and notations

**Preliminary and notations**    We denote $x \in \mathbb{R}^n$ as a clean image and $x'$ as an adversarial example constructed by the adversarial attack such as PGD (Madry et al., 2017). We assume that we have a pretrained classifier $f_{\theta}$, which successfully predicts the label of the given input image $x$. However, $f_{\theta}$ is vulnerable to an attacked image $x'$ in general. Note that we only consider gray-box attack and black-box attack. For gray-box attack case, attacker can only access to the classifier model, not to the defense method, and in black-box attack case, an attacker do not know the information of both the classifier and defense method. The goal of input transformation converting the input data $x$ to $\hat{x} = T(x)$ is to maintain standard accuracy for the clean image $x$, while increasing robust accuracy for the attacked image $x'$.

**FBI-Net**    Blind-spot Network (BSN) is designed for training a denoiser in an unsupervised way and FBI-Net proposed in (Byun et al., 2021) achieves the state-of-the-art blind denoising performance with very short inference time. The reconstruction of $i$-th pixel of a given image $x_i$ of FBI-Net can be denoted as

$$\hat{x}_i = a_0(\boldsymbol{\theta}, C_{k \times k}^{-i})_i = g_{\theta}(x)_i \qquad (1)$$

. Note that $a_0$ is the output of BSN with parameter $\boldsymbol{\theta}$ which receives $x$ as an input image and returns a reconstructed image, $g_{\theta}(x) \in \mathbb{R}^n$. However, it only utilizes the $k \times k$ patch surrounding $x_i$ (which excludes $x_i$, denoted as $C_{k \times k}^{-i}$), as input patch for restoration $\hat{x}_i$.

### 3.2. Iterative smoothing for adversarial defense

The idea of using a variety of input transformation methods for adversarial defense was previously discussed in (Guo et al., 2017a). However, these methods require additional training procedure and still show weakness to strong gray- and black-box attacks. Different from the previous methods, our model introduces an iterative pre-processing of the given input image $\hat{x}_0$, either for attacked and clean images $x'$ and $x$, as in,

$$\hat{x}^{(t+1)} = Clip(S(\hat{x}^{(t)}), 0, 1). \ \ t = 1 \ldots T \qquad (2)$$

Here, $S(\cdot)$ is a smoothing method and the number $T$ denotes the number of iteration. After the iterative pre-processing step, we feed $\hat{x}^{(T)}$ to $f_{\theta}$ for robust classification against adversarial attack. The overall process is shown in Figure 1.

We note that Gaussian or median smoothing can be common solutions, and iterative smoothing is widely used for image denoising (Kumar & Sodhi, 2020). In summary, we firstly apply iterative smoothing to adversarial defense and show that the effectiveness of the iterative smoothing on erasing adversarial perturbations in the attacked samples.

### 3.3. Self-supervised iterative contextual smoothing

**Self-supervised training of FBI-Net**    Different from the usage of BSN in blind image denoising, we propose to train BSN in self-supervised manner by using mean squared error loss function, denoted as:

$$\frac{1}{n}||x - \hat{x}||_2^2 \qquad (3)$$

, where the clean image $x$ is used as the input and target image, and $\hat{x} = g_{\theta}(x)$. One important thing to note is that the loss function can only be trained by BSN, due to the constraints of BSN on the input image $x$.

After training FBI-Net, the restored pixel $\hat{x}_i$ of the given input $x$ can be considered as the *context-to-pixel* smoothed result, because the restoration reconstructs the single pixel $\hat{x}_i$ by utilizing the convolution using the patch information surrounding $x_i$ (but excluding $x_i$) and the trained parameter $\boldsymbol{\theta}$. Figure 2(a) shows the smoothing procedure by FBI-Net.

**Expansion to context-to-context smoothing**    The limitation of *context-to-pixel* restoration is that it only considers a single pixel level restoration. It induces the quality degradation on restoration in non-local features. To overcome this limitation, we propose two simple tensor operations, *patch-to-channel* and *channel-to-patch* operation, to expand the restoration process from *context-to-pixel* to *context-to-context*. As shown in Figure 2(b), the two operations are applied as a pre- and post-processing for FBI-Net respectively. *patch-to-channel* transfers pixels of input image in $P_K \times P_K$ patches to channel-wise pixels and *channel-to-context* exactly operates as a reverse operation of *context-to-channel*.

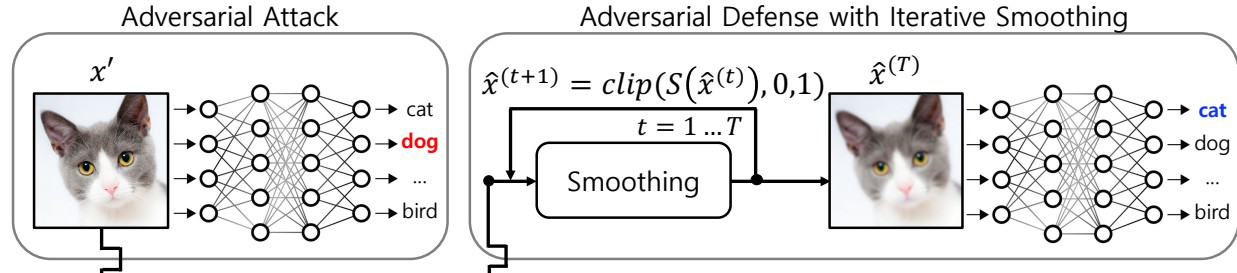

Figure 1. Overall procedure of iterative smoothing for adversarial defense

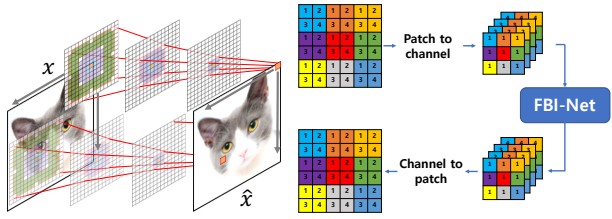

(a) Illustration of smoothing procedure using FBI-Net

(b) Example of context-to-context smoothing for $P_K = 2$.

Figure 2. Illustration and modules of SSICS

By adding these tensor operations, the restoration is simply expanded from *context-to-pixel* to *context-to-context*. For example, Figure 2(b) illustrates that the red color patch (consisting of four pixels) is reconstructed from patch information of eight nearby patches. As a result, it also can be considered as *context-to-context* smoothing using FBI-Net.

**Combining Gaussian smoothing** The level of difficulty is higher to reconstruct *context-to-context* than *context-to-pixel*, we combined Gaussian smoothing with the output of FBI-Net. The restored $\hat{x}_i$ is denoted as,

$$\hat{\boldsymbol{x}}_i = O_r(g_{\boldsymbol{\theta}}(O(\boldsymbol{x})))_i + GS(\boldsymbol{x})_i \quad (4)$$

, where $O$ and $O_r(\cdot)$ denotes *patch-to-channel* and *channel-to-patch* operation, and $g_{\boldsymbol{\theta}}$ is also trained by Equation (3). During training, $g_{\boldsymbol{\theta}}$ learns to reconstruct the residual between a target image and $GS(\boldsymbol{x})$, and then it can be applied to iterative smoothing proposed in Equation (2) as the smoothing function $S(\cdot)$. In conclusion, we call it as Self-Supervised Iterative Contextual Smoothing (SSICS) and experimentally found that, our SSICS not only well removes adversarial perturbations than Gaussian smoothing, but also maintains discriminative features for classification during iterative steps.

## 4. Experimental Results

### 4.1. Experimental settings

**Adversarial attack** Following the experimental setting of (Xie et al., 2019), we evaluate all the baselines and the proposed method using PGD ($L_\infty$ and $L_2$) attack (Madry et al., 2017) on ImageNet validation dataset (Deng et al., 2009). We used ImageNet pretrained ResNet152 (He et al.,

2016) as the classification model for entire experiments. The more details are introduced in SM.

**Evaluation metric and baselines** As an evaluation metric, we report standard accuracy, which is average classification accuracy for clean images, and robust accuracy, which is average classification accuracy for adversarial examples. We select five defense methods as a baseline and FD (Xie et al., 2019), which is the state-of-the-art defense trained by adversarial training, as an empirical upper bound. The detailed description of baselines is introduced in SM.

### 4.2. Gray-box attack

Table 1. Experimental results of adversarial defense against Gray-box PGD attack $L_\infty$ ($\epsilon = 16$, iter = 10). $\sigma$ denotes variance of Gaussian noise for DS and $G_K$ and $P_K$ denote a kernel size for Gaussian smoothing and SSICS respectively. $t$ means the number of steps for iterative smoothing

| Adversarial defense | Standard Accuracy | Robust Accuracy | Average Accuracy | Inference Time (per image) |
|---|---|---|---|---|
| w/o defense | 78.31 | 4.40 | 41.36 | 0.0000 |
| FS | 76.68 | 6.22 | 41.45 | 0.0000 |
| TVM | 69.86 | 17.37 | 43.62 | 0.8962 |
| JPEG | 76.00 | 5.57 | 40.79 | 0.0083 |
| DS ($\sigma = 0.12$) | 65.82 | 36.55 | 51.19 | 1.0300 |
| DS ($\sigma = 0.25$) | 50.50 | 41.87 | 46.19 | 1.0300 |
| DS ($\sigma = 0.5$) | 26.98 | 24.90 | 25.94 | 1.0300 |
| GS ($G_K = 5$, t= 6) | 65.29 | 43.20 | 54.25 | 0.0000 |
| SSICS ($P_K = 0, G_K = 0, t = 7$) | **69.15** | 40.38 | 54.76 | 0.3312 |
| SSICS ($P_K = 2, G_K = 0, t = 7$) | 64.67 | 47.37 | 56.02 | 0.0808 |
| SSICS ($P_K = 2, G_K = 11, t = 7$) | 68.30 | **48.83** | **58.56** | 0.0809 |
| FD | 64.00 | 63.46 | 63.73 | 0.0000 |

**Single gray-box PGD attack** To compare baselines to Gaussian smoothing (GS) and variants of SSICS, we evaluate all the baselines for gray-box attack. we report the best result of each method in Table 1. We generate adversarial examples using gray-box PGD $L_\infty$ attack and evaluate all the methods using the adversarial examples. For training all the variants of SSICS, only the 5% of ImageNet training dataset were used.

From the experimental results, first, we can see that baselines of input transformation, such as, FS (Xu et al., 2017), TVM (Rudin et al., 1992) and JPEG (Dziugaite et al., 2016), could not successfully defend adversarial examples. The observed failures are also demonstrated by previous works (Dong et al., 2020; Guo et al., 2017a). Among them, TVM shows slightly better robust accuracy than others, but it requires long inference time. Second, we observe that DS

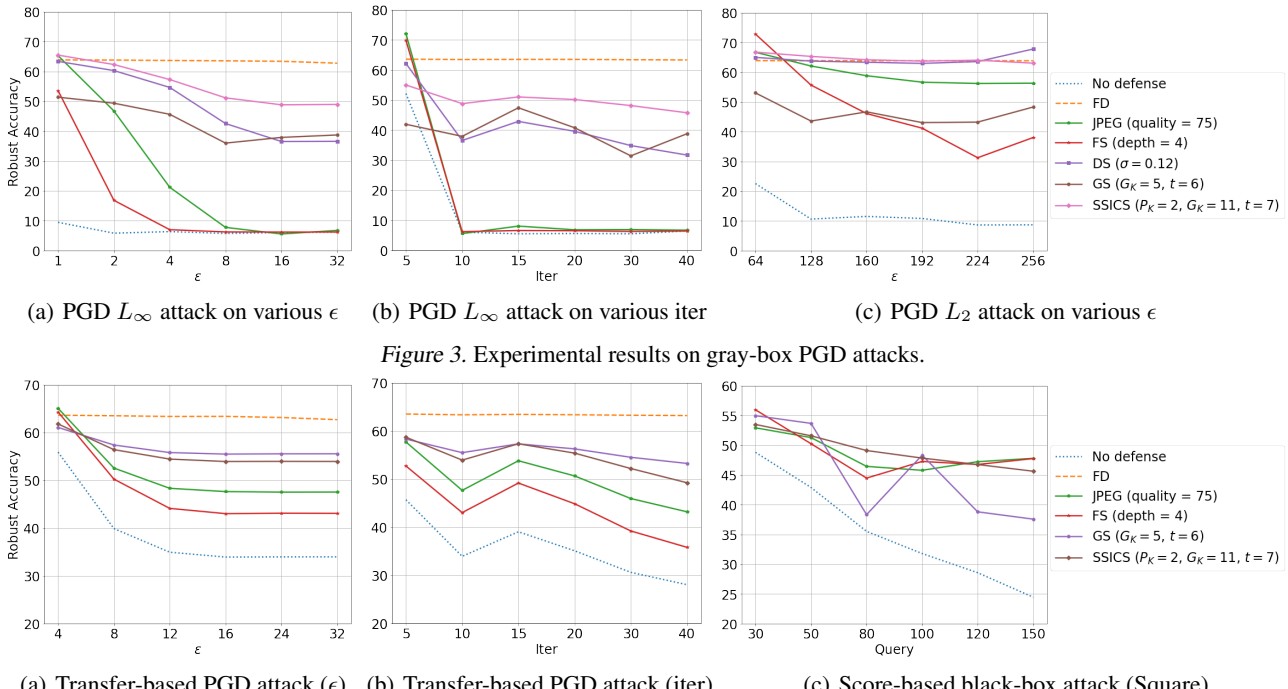

*Figure 3.* Experimental results on gray-box PGD attacks.

(a) Transfer-based PGD attack ($\epsilon$)    (b) Transfer-based PGD attack (iter)    (c) Score-based black-box attack (Square)

*Figure 4.* Experimental results on black-box attacks.

(Salman et al., 2020) performance is highly dependent on the level of Gaussian noise used for randomized smoothing. Even though DS achieves an impressive standard and robust accuracy in $\sigma = 0.12$, it takes long time to defend because of its noise sampling procedure. Finally, we clearly observe that variants of SSICS and GS ($G_K = 5, t = 6$) achieve the impressive result in both standard and robust accuracy compared to all baselines. Epecially, newly proposed SSICS ($C_K = 2, P_K = 11, t = 7$) maintains high standard accuracy, and also well defends adversarial examples with fast inference time. It is worth highlighted in that our average accuracy is comparable to FD and overwhelms the other input transformation baselines, without adversarial training or re-training for the classification model. See SM for the additional results regarding a verification of iterative smoothing.

**Various gray-box PGD attacks** The results in Figure 3 demonstrate that input transformation based methods such as JPEG and FS reveal weakness against strong gray-box attacks, as already shown in (Dong et al., 2020). On the other hand, DS ($\sigma = 0.12$) shows more robust performance to those attacks. Especially, it achieves a superior result in PGD $L_2$ but suffers from slow inference time than other baselines. Note that our SSICS shows consistent robust accuracy for all experiments, and even obtains comparable results to FD.

### 4.3. Black-box attack

We evaluate baselines and SSICS on two types of black-box attack. First, we consider transfer-based PGD attack (Dong et al., 2020) and set DenseNet201 as substitute model

(Huang et al., 2017) to generate adversarial examples. Second, we select Square (Andriushchenko et al., 2020), which is the state-of-the-art score-based black-box attack.

**Transfer-based PGD attack** Figure 4(a) and 4(b) show the experimental results of transfer-based PGD attack on various settings that were already proposed in previous section. Although other baselines show better robust accuracy than previous section, we clearly observe that GS and SSICS are more robust to variants of transfer-based PGD attacks, and only comparable to FD trained by adversarial training.

**Score-based black-box attack** We generate adversarial examples using Square (Andriushchenko et al., 2020) on various queries and evaluate the baselines and SSICS with them, as shown in Figure 4(c). It again shows that JPEG and FS are quite robust to score-based black-box attack than the other type of attacks, as also demonstrated in (Dong et al., 2020), but GS achieves relatively worse results than transfer-based PGD attack. On the other hand, our SSICS maintains competitive robust accuracy for various levels of query compared to JPEG and FS.

## 5. Concluding Remarks

We propose SSICS that extends existing BSN to the context level and combines Gaussian smoothing. As a result, we show simple iterative Gaussian smoothing and SSICS can robustly defend various gray- and black-box attacks without any adversarial training or re-training of a classification model. Our future work is to analyze the reason our SSICS can significantly wash out adversarial perturbation, as well as maintain the discriminative feature for classification.

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

# Self-Supervised Iterative Contextual Smoothing
# for Efficient Adversarial Defense against Gray- and Black-Box Attack

**Sungmin Cha** [1 2]  **Naeun Ko** [3]  **Youngjoon Yoo** [2 3]  **Taesup Moon** [1]

## 1. Additional Related Works

**Blind-spot networks for blind image denoising**  The goal of blind image denoising is to train a denoiser network using only a single noisy image. The most dominant direction in blind image denoising is a blind-spot network (BSN) method. BSN restores each pixel by only the given context around the pixel, excluding the target pixel. After the success of BSN, many modified methods (Krull et al., 2019; Cha & Moon, 2018; 2019; Laine et al., 2019; Wu et al., 2020) have been proposed. However, these methods commonly share the problem of time and cost inefficiency because of its inefficient architecture. Recently, (Byun et al., 2021) proposes the newly devised BSN, called as FBI-Net, which achieves the fastest inference time and the lowest GPU memory usage than existing all baselines.

## 2. The Details on Experimental Settings

**Baselines**  We select four input transformation based adversarial methods as baseline, abbreviated as FS(Xu et al., 2017), SS (Xu et al., 2017), JPEG (Dziugaite et al., 2016) and TVM (Rudin et al., 1992) proposed in (Guo et al., 2017). The significant difference to (Guo et al., 2017) is that we do not re-train the classifier on a given transformation method for defense in all experiments. Also, we evaluate DS (Salman et al., 2020) as a representative of certified defense because it does not require any pre-training of classifier, like our method. We set the number of sampling for DS as $n = 100$. Finally, we select FD (Xie et al., 2019), which denotes an adversarially trained model, as an upper bound of defense. We conducted experiments by implementing code in (Ding et al., 2019; Nicolae et al., 2018; Kim, 2020), and downloaded weights of (Salman et al., 2020) and (Xie et al., 2019) from their official website.

[1]Department of Electrical and Computer Engineering, Seoul National University, South Korea [2]NAVER AI Lab, South Korea [3]Face, NAVER Clova, South Korea. Correspondence to: Taesup Moon <tsmoon@snu.ac.kr>.

*Accepted by the ICML 2021 workshop on A Blessing in Disguise: The Prospects and Perils of Adversarial Machine Learning.* Copyright 2021 by the author(s).

**Hyperparameter settings for PGD attack**  We used a specified $\epsilon$ and iter (number of steps for gradient update) for all experiments in the manuscript. For $\alpha$ (the strength of update for each iteration), we set $\alpha = 16/255$. when the number of steps (iter) is less than or equal to 10, $\alpha = \epsilon/255$. when the number of steps (iter) is greater than 10.

**Controlling $\sigma$ for Gaussian smoothing**  For Gaussian smoothing (GS) (and also GS in SSICS), we set $\sigma = G_K - 1/6$ and only controls the kernel size $G_K$ as a hyperparameter.

## 3. Additional Experimental Results

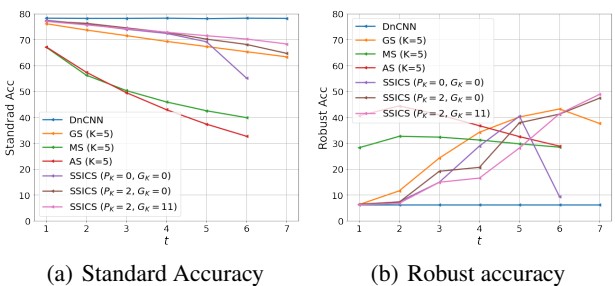

(a) Standard Accuracy          (b) Robust accuracy

*Figure 1.* Verification of iterative context smoothing

**Verifying iterative context smoothing**  We verify the advantage of iterative context smoothing for adversarial defense as shown in Figure 1. For that, we generate adversarial examples using gray-box PGD $L_\infty$ ($\epsilon = 16$, iters=10) attack and evaluate three types of smoothing, such as Gaussian, average and median smoothing with the kernel size of 5 ($K = 5$), and SSICS. Note that we trained DnCNN (Zhang et al., 2017) and all variants of SSICS, in a self-supervised way, using only $5\%$ of whole images in ImageNet training dataset. From the results, we could find out that, firstly, iterative context smoothing with GS ($K = 5$) works better than other types of smoothings, such as AS and MS, in both standrand and robust accuracy. Especially, when we iterate GS ($K = 5$) for 6 times, it achieves a superior accuracy in both cases. Secondly, SSICS ($P_K = 0, G_K = 0$), which is equal to the original FBI-Net (Byun et al., 2021), obtain a better result at $t = 5$ than GS ($K = 5$), however, it is sensitively

degraded from $t = 6$. Thirdly, when we extend SSICS to *context-to-context* with $P_K = 2$, there is no degradation and it achieves the most superior standard and robust accuracy even than GS ($K = 5$), in addition, if GS is used together, SSICS ($P_K = 2, G_K = 11$) achieves a slight increase in both standard and robust accuracy at $t = 7$. Finally, the result of DnCNN, which is trained by Equation (3) in the manuscript, demonstrates that the restoration model based on a pure convolutional neural network cannot be used for SSICS. This is because there is no constraint for $x_i$, which is discussed in Section 3.1 in the manuscript. Therefore, DnCNN trained by self-supervised learning perfectly returns the given input image directly, as a result, it achieves robust accuracy it had when defense was not applied.

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
