# OpenReview forum: "Self-Supervised Iterative Contextual Smoothing for Efficient Adversarial Defense against Gray- and Black-Box Attack"
_ICML.cc/2021/Workshop/AML — ICML 2021 Workshop AML Poster_

### Official Review · Reviewer_Edws · 2021-06-19
**The paper proposed Self-Supervised Iterative Contextual Smoothing(SSICS), which is an adversarial defense method based on input transformation.**

**Rating:** Accept
**Confidence:** 3

**Review:**

The paper considered input transformation based defense method against gray- and black-box attacks and proposed Self-Supervised Iterative Contextual Smoothing (SSICS), which aims to iteratively reconstruct the original discriminative features from the Gaussian-smoothed image in a context-adaptive way. There are three main techinques in SSICS: self-supervised training of FBI-Net, context-to-context smoothing and Combining Gaussian smoothing. The experiments on ImageNet show the effectiveness of SSICS against gray- and black-box attack. However, the novelty is not such remarkable. What’s more, the theoretical guarantee of the robustness of SSICS may be relevant to Random Smoothing since they both use Gaussian Smoothing.

---

### Decision · Program_Chairs · 2021-06-21

**Decision:**

Accept (Poster)

**Comment:**

This paper considered input transformation defense against gray- and black-box attacks and proposed Self-Supervised Iterative Contextual Smoothing (SSICS). The techniques are solid and the experiments show the effectiveness.